# Analyzing Best-Response Dynamics for Cooperation in Markov Potential Games

**Dingyang Chen***  *dingyangchen118@gmail.com*
*Amazon*

**Xiaoling Zeng**  *xzeng4@wpi.edu*
*Worcester Polytechnic Institute*

**Thinh T. Doan**  *thinhdoan@utexas.edu*
*University of Texas at Austin*

**Qi Zhang**  *qzhang9@wpi.edu*
*Worcester Polytechnic Institute*

**Reviewed on OpenReview:** *https://openreview.net/forum?id=klFSzxt4MC*

## Abstract

Simultaneous gradient updates are widely used in multi-agent learning. However, this method introduces non-stationarity from the perspective of each agent due to the co-evolution of other agents' policies. To address this issue, we consider best-response dynamics, where only one agent updates its policy at a time. We theoretically show that with best-response dynamics, convergence results from single-agent reinforcement learning extend to Markov potential games (MPGs). Moreover, building on the concept of price of anarchy and smoothness from normal-form games, we aim to find policies in MPGs that achieve optimal cooperation and provide the first known suboptimality guarantees for policy gradient variants under the best-response dynamics. Empirical results demonstrate that the best-response dynamics significantly improves cooperation across policy gradient variants in classic and more complex games.

## 1 INTRODUCTION

The framework of multi-agent sequential decision making is often formulated as (variants of) Markov games (MGs) (Shapley, 1953). This paper focuses on *Markov potential games* (MPGs) (Macua et al., 2018; Leonardos et al., 2021; Zhang et al., 2021), a subclass of MGs that is extended from the notion of (normal-form) potential game (Monderer & Shapley, 1996; Roughgarden, 2016) and incorporates as an important special case the fully cooperative MGs where all agents share the same reward, making them applicable to real-world scenarios like multi-robot coordination (Corke et al., 2005), traffic control (Chu et al., 2019), and power grid management (Callaway & Hiskens, 2010).

Nash policy is perhaps the most well-known MG solution concept, where every agent selects its actions independently of any other given the state and reaches an equilibrium by playing a best response to all others. MPGs often model problems where outcomes of high social welfare, measured by the sum of all agents' values, are most desirable. In these scenarios, the solution concept of the Nash policy is inadequate as it alone does not characterize the quality of agents' cooperation at equilibrium. The goal of this paper is to *find Nash policies in MPGs that are near-optimal*, where the optimality notion is measured by the sum of all agents' values.

---

*Work does not relate to position at Amazon.

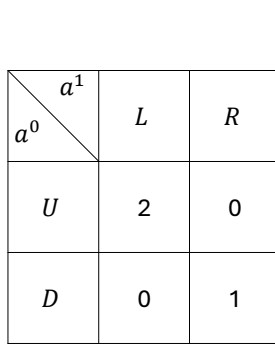
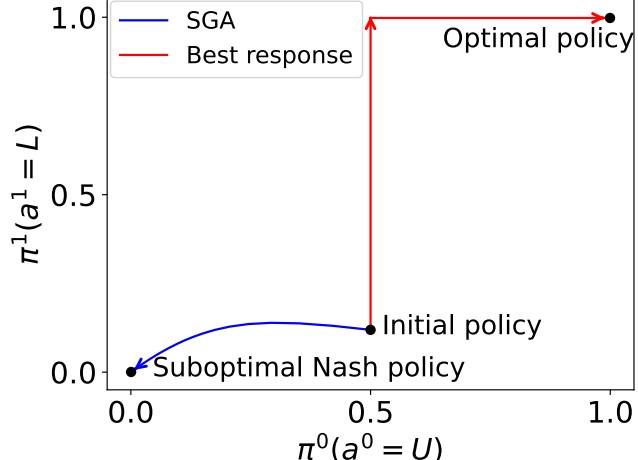

Figure 1: *Left:* Payoff matrix for a two-agent matrix game with binary actions $\{U, D\}$ and $\{L, R\}$. Two Nash equilibria exist: 1) optimal $(U, L)$, and 2) suboptimal $(D, R)$. *Right:* Comparison of simultaneous vs. best-response updates.

As MPG is structured by a potential function, where the change in any agent's value caused by its unilateral policy change can be quantified by the change in the potential, prior work has been focused on invoking Simultaneous Gradient Ascent (SGA) to find a Nash policy, where all agents concurrently update their individual policy parameter in the direction of the gradient on the potential (Monderer & Shapley, 1996; Roughgarden, 2016; Balduzzi et al., 2018). It is unclear, however, whether the found Nash is likely to be near-optimal or not. In this paper, we theoretically and empirically compare SGA with an alternative dynamics, known as the Best-Response (BR) dynamics, and particularly focus on the *unilateral BR* variant where only one agent updates its policy at a time to (approximately) optimize its value while fixing all other agents' policies. Therefore, Nash found by unilateral BR is likely to be drastically different from those by SGA. Figure 1 illustrates the benefit of BR over SGA in a matrix game, where, starting from the same initialization, BR converges to the better Nash while SGA to the worse. Interestingly, our experiments in Section 7 will show that BR converges to the better Nash more often than SGA starting from random initializations. This motivates us to carefully compare the two dynamics.

Specifically, our claims and contributions are:

1) Section 4 develops a unilateral BR dynamics for MPGs where one agent is selected to approximately compute its best response. This approach contrasts with existing BR variants that rely on immediate optimal responses, which are impractical to implement.

2) Section 5.1 establishes a convergence rate of unilateral BR to Nash when the agent that would maximally improve the potential is selected, known as the maximum-gain criterion. This convergence rate is first for BR to match that of SGA. Moreover, the section establishes asymptotical convergence guarantees of unilateral BR for both the maximum-gain and the more practical round-robin agent-selection criteria.

3) Section 5.2 considers a MPG smoothness condition by which near-Nash policies are also near-optimal, leading to Theorem 3 that confirms maximum-gain BR matches SGA in terms of both Nash and optimality guarantees *at convergence* and Theorem 4 that suggests the advantage of BR over alternatives like SGA for deriving near-optimal policies *during the updates*.

4) Section 6 incorporates unilateral BR into decentralized deep multi-agent reinforcement learning (MARL) algorithms, where all restrictive assumptions required in the theoretical results are relaxed. Specifically, for practical implementation, maximum-gain is replaced with the round-robin criterion. Experiments in Section 7 show the effectiveness of the BR-based methods over their SGA counterparts in improving cooperation for a wide range of problems, from stateless matrix games to stateful and high-dimensional ones.

## 2 RELATED WORK

**Single-agent policy gradient convergence.** Agarwal et al. (2019) first established the policy gradient convergence to global optima in the single-agent setting under tabular softmax parameterization, specifically, asymptotic convergence of policy gradient ascent, finite-time convergence with log barrier regularization, and finite-time convergence with natural policy gradient. Mei et al. (2020) later established finite-time convergence of (regularized) policy gradient ascent under tabular softmax parameterization, with a convergence rate depending on a problem-specific variable. This problem-specific variable in some sense is necessary, as Li et al. (2021) show that the softmax policy gradient can take exponential time to converge.

**Policy gradient convergence in MPGs.** Extending Agarwal et al. (2019) from the single-agent setting, Leonardos et al. (2021) and Zhang et al. (2021) both established finite-time convergence of projected gradient ascent under tabular direct parameterization to near-Nash policies in MPGs. Fox et al. (2022) established the asymptotic convergence of the natural policy gradient to Nash policies in MPGs. Zhang et al. (2022b) established finite-time convergence of gradient ascent under tabular softmax parameterization to near-Nash policies in MPGs. Sun et al. (2023) has shown that the policy gradient method converges at a rate $O(1/\epsilon)$, which is the same as the one in the single-agent setting. However, these prior works do not quantify the performance of the convergent Nash policies.

**Price of anarchy and welfare guarantees.** Mirrokni & Vetta (2004) initiated the discussion on suboptimality beyond Nash equilibria through the price of anarchy (PoA). Roughgarden (2015) defined smoothness for normal-form games and established PoA bounds for near-Nash equilibria, as well as for maximum-gain best-response dynamics in smooth potential games. A Nash equilibrium only captures unilateral stability and does not by itself quantify welfare quality, which motivates PoA and smoothness beyond pure convergence to Nash. While these tools are well developed in normal-form games (Roughgarden, 2016), there is much less work in Markov games. Zhang et al. (2023) study finite-horizon Markov games with decoupled dynamics under local policies, introduce a notion of smooth Markov games, and use it to bound the price of anarchy; for the MPG case, they also develop a distributed multi-agent soft policy iteration algorithm with Nash-convergence and sample-complexity guarantees. In contrast, our work studies discounted MPGs and uses smoothness not only to relate near-Nash policies to near-optimality, but also to derive welfare/suboptimality guarantees for policies generated by unilateral BR dynamics.

**BR dynamics.** Best-response (BR) dynamics is a foundational topic extensively studied in game theory (Gilboa & Matsui, 1991; Matsui, 1992; Hofbauer & Sigmund, 2003; Marden, 2012; Cortés & Martínez, 2015; Swenson et al., 2018). These studies focus on BR dynamics with immediate optimal responses, where agents concurrently compute optimal strategies assuming fixed policies for others. While this approach offers strong theoretical guarantees, it faces two key challenges: 1) non-stationarity due to simultaneous updates, and 2) the high computational cost of solving single-agent RL problems (where each agent treats others as part of the environment) to compute optimal responses. In this work, we study an alternative that addresses these challenges by updating one agent at a time to avoid non-stationarity and approximating the optimal response through a fixed number of gradient-based updates, thereby reducing computational overhead. This method is also simpler to implement in deep MARL compared to the existing prior works on teammate modeling (Tian et al., 2019; Jaques et al., 2019; Papoudakis et al., 2021; Yuan et al., 2022; Wang et al., 2022; Aghajohari et al., 2024; Jiang et al., 2024), which addressed the non-stationarity by estimating other agents' actions and computing BR-related statistics, such as critic values, for concurrent updates. However, these approaches often suffer from estimation errors which exacerbate non-stationarity. In contrast, our alternative updates one agent at a time based on simple criteria, such as cyclic order, turning each update into a single-agent problem without requiring estimation of others' behaviors. We emphasize that unilateral or turn-taking updates are not new as a generic optimization heuristic; related forms of alternating or sequential updates have also appeared in robust RL and adversarial training (Pinto et al., 2017; Zhang et al., 2022a). Our contribution is instead to study this update structure in discounted Markov potential games and analyze it through Nash-convergence and smoothness-based welfare guarantees. There are additional related works on BR and sequential-update dynamics in potential and Markov games (Swenson et al., 2018; Marden, 2012; Maheshwari et al., 2022; Song et al., 2021; Guo et al., 2023); a more detailed comparison with our work is provided in Appendix A.

## 3 PRELIMINARIES

We consider a Markov game (MG) $\langle \mathcal{N}, \mathcal{S}, \mathcal{A}, P, \vec{r}, \gamma, \mu \rangle$ with $N$ agents indexed by $i \in \mathcal{N} = \{1, \ldots, N\}$, state space $\mathcal{S}$, action space $\mathcal{A} = \mathcal{A}^1 \times \cdots \times \mathcal{A}^N$, transition function $P : \mathcal{S} \times \mathcal{A} \to \Delta(\mathcal{S})$, reward functions $\vec{r} = \{r^i\}_{i \in \mathcal{N}}$ with $r^i : \mathcal{S} \times \mathcal{A} \to [0, \infty)$ being nonnegative for each $i \in \mathcal{N}$, discount factor $\gamma \in [0, 1)$, and initial state distribution $\mu \in \Delta(\mathcal{S})$. Our theoretical results in Sections 4 and 5 assume full observability, i.e., each agent observes the state $s \in \mathcal{S}$, and this assumption will be relaxed for our practical algorithm and empirical study in Sections 6 and 7. Under full observability, we consider *product policies*, $\pi : \mathcal{S} \to \times_{i \in \mathcal{N}} \Delta(\mathcal{A}^i)$, that is factored as the product of individual policies $\pi^i : \mathcal{S} \to \Delta(\mathcal{A}^i)$, $\pi(a|s) = \prod_{i \in \mathcal{N}} \pi^i(a^i|s)$. Define the discounted return for agent $i$ from time step $t$ as $G_t^i = \sum_{l=0}^{\infty} \gamma^l r_{t+l}^i$, where $r_t^i := r^i(s_t, a_t)$ is the reward at time step $t$ for agent $i$. For agent $i$, product policy $\pi = (\pi^1, \ldots, \pi^N)$ induces a value function defined as $V_\pi^i(s_t) = \mathbb{E}_{s_{t+1:\infty}, a_{t:\infty} \sim \pi}[G_t^i|s_t]$, action-value function $Q_\pi^i(s_t, a_t) = \mathbb{E}_{s_{t+1:\infty}, a_{t+1:\infty} \sim \pi}[G_t^i|s_t, a_t]$, and advantage function $A_\pi^i(s_t, a_t) = Q_\pi^i(s_t, a_t) - V_\pi^i(s_t)$. Following policy $\pi$, agent $i$'s cumulative reward starting from $s_0 \sim \mu$ is denoted as $V_\pi^i(\mu) := \mathbb{E}_{s_0 \sim \mu}[V_\pi^i(s_0)]$. It will be useful to define the (unnormalized) *discounted state visitation measure* by following policy $\pi$ after starting in $s_0 \sim \mu$: $d_\mu^\pi(s) := \mathbb{E}_{s_0 \sim \mu}\left[\sum_{t=0}^{\infty} \gamma^t \mathrm{Pr}^\pi(s_t = s|s_0)\right]$ where $\mathrm{Pr}^\pi(s_t = s|s_0)$ is the probability that $s_t = s$ after starting at state $s_0$ and following $\pi$ thereafter.

We focus on the MG subclass of *Markov potential games* (MPGs) (Macua et al., 2018; Leonardos et al., 2021; Zhang et al., 2021) as defined in Definition 1, which is extended from the notion of (normal-form) potential game and also incorporates as a special case the fully cooperative MGs where all agents share the same reward function.

**Definition 1** (Markov potential game). A Markov game is called a *Markov potential game* (MPG) if there exists a potential function $\phi : \mathcal{S} \times \mathcal{A} \to \mathbb{R}$ such that for any agent $i$, any pair of product policies $(\pi^i, \pi^{-i}), (\bar{\pi}^i, \pi^{-i})$, and any state $s$ to be considered as the initial state, we have: letting $\phi_t := \phi(s_t, a_t)$, the *total potential function* $\Phi_\pi(s) := \mathbb{E}_{s_0 = s, \ s_{1:\infty}, a_{0:\infty} \sim \pi}\left[\sum_{t=0}^{\infty} \gamma^t \phi(s_t, a_t)\right]$,

$$V_{\bar{\pi}^i, \pi^{-i}}^i(s) - V_{\pi^i, \pi^{-i}}^i(s) = \Phi_{\bar{\pi}^i, \pi^{-i}}(s) - \Phi_{\pi^i, \pi^{-i}}(s) \tag{1}$$

We also define $\Phi_\pi(\mu) := \mathbb{E}_{s_0 \sim \mu}[\Phi_\pi(s_0)]$. In the tabular setting, since $\phi : \mathcal{S} \times \mathcal{A} \to \mathbb{R}$ is defined on a finite domain, $\phi$ is automatically bounded. Hence the discounted total potential $\Phi_\pi(s)$ is also uniformly bounded over all $s, \pi$, i.e., there exist constants $\Phi_{\min}, \Phi_{\max}$ such that

$$\Phi_{\min} \leq \Phi_\pi(s) \leq \Phi_{\max}, \qquad \forall s, \pi.$$

We focus on the solution concepts of Nash policy and $\epsilon$-best response:

**Definition 2** ($\epsilon$-Nash policy and $\epsilon$-best response). The *Nash-gap* of a product policy $\pi$ is defined as

$$\texttt{Nash-gap}^i(\pi) := \max_{\bar{\pi}^i} V_{\bar{\pi}^i, \pi^{-i}}^i(\mu) - V_\pi^i(\mu),$$

$$\texttt{Nash-gap}(\pi) := \max_i \texttt{Nash-gap}^i(\pi).$$

Policy $\pi$ is $\epsilon$-*Nash* if $\texttt{Nash-gap}(\pi) \leq \epsilon$. For agent $i$, $\pi^i$ is its $\epsilon$-best response if $\texttt{Nash-gap}^i(\pi) \leq \epsilon$.

While the notions of Nash and best response define a equilibrium where no individual agents would deviate unilaterally, we are also interested in the cooperation quality of a policy defined by an optimality notion that lower bounds the ratio between the values summed over all agents and the largest summed values achieved by any product policy. Note that it is sound to consider the ratio because we have required the rewards to be nonnegative.

**Definition 3** ($\delta$-optimal policy). Policy $\pi$ is $\delta$-*optimal* if $\frac{V_\pi(\mu)}{\max_{\bar{\pi}} V_{\bar{\pi}}(\mu)} \geq \delta$ where $V_\pi(s) := \sum_i V_\pi^i(s)$.

## 4 DYNAMICS IN MPGs

We consider the product policy parameterized per agent. Denoting agent $i$'s policy parameter as $\theta^i$, the product policy is therefore parameterized as $\pi_\theta = (\pi_{\theta^1}^1, \ldots, \pi_{\theta^N}^N)$ where $\theta = (\theta^1, \ldots, \theta^N)$ consists of all agents' policy parameters. We will abbreviate $\Phi_{\pi_\theta}$ and $V_{\pi_\theta}^i$ as $\Phi_\theta$ and $V_\theta^i$, respectively. By *dynamics*, we

mean the process in which all agents initialize their policy parameters as $\theta_0 = (\theta_0^1, \ldots, \theta_0^N)$ and update them at iterations $t = 0, 1, \ldots$, generating a sequence of product policies parameterized by $(\theta_t)_{t \geq 0}$. We will focus on two classic families of dynamics, the *Simultaneous Gradient Ascent* and the *Best-Response* dynamics.

**Simultaneous Gradient Ascent (SGA).** In SGA (Singh et al., 2000; Zhang & Lesser, 2010; Balduzzi et al., 2018; Wang et al., 2019; Schäfer & Anandkumar, 2019), every agent simultaneously updates its policy parameter by taking a direction with a certain learning rate:

$$\text{(SGA)} \qquad \theta_{t+1}^i = \theta_t^i + \eta g^i(\theta_t) \quad \forall\, i \tag{2}$$

where we are restricted to 1) update direction $g^i(\theta_t)$ being dependent only on the policy at the current iteration and 2) learning rate $\eta$ being a constant.

The update direction is often chosen to improve the agent's value. We will consider two common choices for the update direction that aims to improve the agent's value: the policy gradient (PG)

$$\text{(PG)} \qquad g^i(\theta_t) = \nabla_{\theta^i} V_{\theta_t}^i(\mu) = \nabla_{\theta^i} \Phi_{\theta_t}(\mu) \quad \forall\, i \tag{3}$$

where the second equality is due to the potential function structure Eq. equation 1, which implies $\nabla_{\theta^i} V_\theta^i(s) = \nabla_{\theta^i} \Phi_\theta(s)$ for any state $s$, and the natural policy gradient (NPG)

$$\text{(NPG)} \qquad g^i(\theta_t) = (F_{\theta_t}^i)^\dagger \nabla_{\theta^i} V_{\theta_t}^i(\mu) = (F_{\theta_t}^i)^\dagger \nabla_{\theta^i} \Phi_{\theta_t}(\mu) \quad \forall\, i \tag{4}$$

where $X^\dagger$ is the Moore–Penrose inverse of a matrix $X$ and $F_\theta^i$ is the Fisher information matrix for agent $i$ under product policy $\pi_\theta$:

$$F_\theta^i = \mathop{\mathbb{E}}_{s \sim d_\mu^{\pi_\theta}, a^i \sim \pi^i(s)} \left[ \nabla_{\theta^i} \log \pi_{\theta^i}^i(a^i|s) \nabla_{\theta^i} \log \pi_{\theta^i}^i(a^i|s)^\top \right].$$

**Best response (approximate and unilateral).** In the Best-Response (BR) dynamics (Cesa-Bianchi & Lugosi, 2006; Roughgarden, 2016; Rajeswaran et al., 2020), instead of taking an incremental improvement step, an agent optimizes its policy parameter assuming all other agents are fixed. In practice, the best response can be approximated using a large number of gradient steps. Formally, we set a constant number of $K$ iterations for an agent to approximate its best response starting at iteration $t = lK$ for $l = 0, 1, 2, \ldots$. For some $t = lK$, an agent $i$ sets $\tilde{\theta}_{t,0}^i = \theta_t^i$ and performs iterations $k = 1, \ldots, K$ to approximate its best response:

$$\text{(Approximate BR)} \qquad \tilde{\theta}_{t,k}^i = \tilde{\theta}_{t,k-1}^i + \eta g^i\left( \tilde{\theta}_{t,k-1}^{i,-i} \right) \tag{5}$$

where $\tilde{\theta}_{t,k-1}^{i,-i} = \left\{ \tilde{\theta}_{t,k-1}^i, \theta_t^{-i} \right\}$ includes agent $i$'s parameter during the $K$ iterations while fixing the parameters of other agents at $t = lK$. Instead of updating all agents' parameters to their (approximate) best response, for every $K$ iterations indexed by $l$, we select only one agent $i_l^* \in \mathcal{N}$ to perform its BR update, resulting in what is referred to as unilateral BR (Roughgarden, 2016):

$$\text{(Unilateral BR)} \qquad \theta_{t+k}^{i_l^*} = \tilde{\theta}_{t,k}^{i_l^*}, \; \theta_{t+k}^i = \theta_t^i \text{ for } i \neq i_l^*. \tag{6}$$

Note this BR variant is both unilateral and approximate, which is the variant this paper focuses on. We consider two criteria to determine $i_l^*$. *Maximum gain* chooses the agent that would maximally improve the total potential:

$$i_l^* = \arg\max_i \left( \Phi_{\tilde{\theta}_{t,K}^{i,-i}}(\mu) - \Phi_{\theta_t}(\mu) \right). \tag{7}$$

*Round robin* updates the agents in a cyclic manner:

$$i_l^* = 1 + (l \mod N). \tag{8}$$

## 5 CONVERGENCE ANALYSIS

It is known that SGA converges to Nash policies (Zhang et al., 2022b), yet the convergence guarantees of BR in MPGs have been understudied. In this section, we establish guarantees for approximate BR with maximum-gain updates converging to Nash policies, with a rate that matches that of SGA. Further, we link near-Nash policies to near-optimal policies in MPGs with certain smoothness conditions, which enables us provide a suboptimality guarantee that is unique to maximum-gain BR. In this section, we make the following assumption on the discounted state visitation distribution.

**Assumption 1.** $d_\mu^\pi(s) > 0$ for any $\pi$ and $s$.

Assumption 1 is standard in prior work (Agarwal et al., 2019; Zhang et al., 2021). A sufficient condition is that $\mu(s) > 0$ for all $s \in \mathcal{S}$, ensuring all states are reachable.

### 5.1 Convergence of BR to Nash

We first establish guarantees for maximum-gain BR converging to Nash policies. This result extends the single-agent counterparts with the following reasoning. Suppose we aim to converge to a $\epsilon$-Nash policy. We can set $K$ large enough, such that every agent's inner loop update (indexed by $k$ in Eq. equation 5) achieves at least $\frac{\epsilon}{2}$-best-response. Therefore, if no agent's improvement in their local value or, equivalently, in the total potential function as computed in Eq. equation 5 is larger than $\frac{\epsilon}{2}$, then the product policy is already an $\epsilon$-Nash policy; otherwise, we can significantly improve the total potential function such that the total number updates can be bounded. This is formally stated below:

**Lemma 1** (Proof in Appendix B.1). *Suppose for some $\epsilon, \eta, p > 0$, all agents achieve at least $\frac{\epsilon}{2}$-best response within $K = O\left(\frac{1}{\epsilon^p}\right)$ iterations of Eq. equation 5 . Then, the maximum-gain approximate BR of Eq. (5,6,7) converges to an $\epsilon$-Nash policy $\theta_T$ with $T = O\left(\frac{\Phi_{\max} - \Phi_{\min}}{\epsilon^{p+1}}\right)$.*

Lemma 1 is applicable to any policy parameterization. In the following, we focus on the specific case of tabular softmax policy parameterization. Individual policies $(\pi^1, \ldots, \pi^N)$ are independently parameterized in a softmax tabular form from the global state. For each agent $i$, its policy is parameterized by $\theta^i = \left\{\theta_{s,a^i}^i \in \mathbb{R} : s \in \mathcal{S}, a^i \in \mathcal{A}^i\right\}$ as:

$$\pi_{\theta^i}^i(a^i|s) = \exp\left(\theta_{s,a^i}^i\right) \Big/ \sum_{\bar{a}^i \in \mathcal{A}^i} \exp\left(\theta_{s,\bar{a}^i}^i\right). \tag{9}$$

Prior work has established convergence rates for single-agent tabular softmax policy parameterization, particularly for both PG (Mei et al., 2020) and NPG (Agarwal et al., 2019). For PG, the rate is given by $K = O\left(\frac{1}{c(1-\gamma)^6\epsilon}\right)$, where $c$ is a constant dependent on the single-agent Markov decision process (MDP) and the initialization. For simplicity, we abuse the notation and assume the existence of a small constant $c$ that provides a lower bound for $c$ across any MDP reduced from MPGs by the best-response dynamics. For NPG, the convergence rate is $K = O\left(\frac{1}{(1-\gamma)^2\epsilon}\right)$. These single-agent convergence results can be utilized by Lemma 1 to establish convergence rates for MPGs:

**Theorem 1** (Proof in Appendix B.2). *With $K \geq \frac{2}{c(1-\gamma)^6\epsilon}$ iterations of Eq. equation 5 of PG update equation 3 and $\eta = (1-\gamma)^3/8$, the maximum-gain approximate BR of Eq. (5,6,7) converges to an $\epsilon$-Nash policy $\theta_T$ with $T = O\left(\frac{\Phi_{\max} - \Phi_{\min}}{c(1-\gamma)^6\epsilon^2}\right)$. With $K \geq \frac{4}{(1-\gamma)^2\epsilon}$ iterations of Eq. equation 5 of NPG update (cf. Eq. equation 4) with $\eta = (1-\gamma^2)\log|\mathcal{A}|$, the maximum-gain approximate BR of Eq. (5,6,7) converges to an $\epsilon$-Nash policy $\theta_T$ with $T = O\left(\frac{\Phi_{\max} - \Phi_{\min}}{(1-\gamma)^2\epsilon^2}\right)$.*

Theorem 1 establishes the same convergence rate for both maximum-gain best-response PG and NPG as for PG- and NPG-based SGA in Zhang et al. (2022b), which relies on the strong assumption that each agent's policy converges asymptotically. In practice, maximum-gain is difficult to realize: it requires centralized per-agent gain evaluation, accurate value/advantage estimates, and tight synchronization; it also scales poorly as the number of agents grows. Theorem 1 further requires a large $K$ to guarantee sufficient inner-loop gain, which is often impractical. Below we show that approximate PG-BR converges asymptotically for any fixed $K$, under both maximum-gain equation 7 and round-robin equation 8.

**Theorem 2** (Proof in Appendix B.3). *With an arbitrary number of inner-loop iterations $K$ and a sufficiently small learning rate $\eta$ (e.g., $\eta = (1-\gamma)^3/8$), the PG-based unilateral approximate BR dynamics of Eq. (3,5,6) converges asymptotically to a Nash policy for both maximum-gain equation 7 and round-robin equation 8.*

Our proof uses monotonic improvement to show that, under both greedy maximum-gain and cyclic round-robin selection, the potential function converges. This convergence implies a zero-gradient state, hence a Nash equilibrium by the single-agent asymptotic-optimality result of Agarwal et al. (2019), which applies here because best response reduces the MPG to a single-agent MDP for the updating agent.

### 5.2 Suboptimality in Smooth MPGs

**Linking Nash to optimality via smoothness.** We ultimately aim to find near-optimal policies (cf. Definition 3). Inspired by prior work in normal-form games (Roughgarden, 2016), Definition 4 extends the notion of *smoothness* in normal-form games to Markov games:

**Definition 4** (Smooth MG). A Markov game is $(\alpha, \beta)$-smooth if $\sum_{i \in \mathcal{N}} V^i_{\bar{\pi}^i, \pi^{-i}}(s) \geq \alpha V_{\bar{\pi}}(s) - \beta V_{\pi}(s)$ for any $s$ and any pair of product policies $\pi, \bar{\pi}$, where $\alpha, \beta$ are nonnegative.

Intuitively, in a smooth Markov game, the externality imposed by one agent on the value of the others is limited. Therefore, we conjecture that a sufficient condition is that both the transition and reward functions of the Markov game are "smooth", which are verified as a proposition in Appendix B.4. In Section 7, we will give two examples of smooth MGs that are involved in our experiments.

As an extension from Roughgarden (2016), we have Theorem 3 stating that near-Nash policies in smooth MGs are near-optimal. To ease presentation, we describe the result for $\epsilon$-*ratio-Nash* policies as the near-Nash notion:

**Definition 5** ($\epsilon$-ratio-Nash policy). Policy $\pi = (\pi^1, \ldots, \pi^N)$ is an $\epsilon$-*ratio-Nash policy* if, for any agent $i$, $\max_{\bar{\pi}^i} V^i_{\bar{\pi}^i, \pi^{-i}}(\mu) \leq \frac{1}{1-\epsilon} V^i_\pi(\mu)$.

**Theorem 3** (Proof in Appendix B.5). *In an $(\alpha, \beta)$-smooth MG, any $\epsilon$-ratio-Nash policy is $\frac{(1-\epsilon)\alpha}{1+(1-\epsilon)\beta}$-optimal.*

**Suboptimality guarantees for maximum-gain BR in smooth MPGs.** With Theorem 3 in place, we have established a guarantee for maximum-gain BR to converge to near-optimal policies: maximum-gain BR converges to near-Nash policies by Theorems 1 and 2, and if the MPG is smooth then the policies at convergence are also near-optimal by Theorem 3 Note such an optimality guarantee applies to any dynamics that finds near-Nash policies. It does not distinguish the BR from SGA because, at convergence, the suboptimality guarantee is irrespective of the dynamics. Also, the rate of convergence to near-Nash is of the same order for the two dynamics, both at $O(1/\epsilon^2)$.

We now establish another suboptimality guarantee that is unique to BR, which is inspired by the counterpart in smooth (normal-form) potential games (Roughgarden, 2015) that bounds the number of highly suboptimal policies generated from the *maximum-gain $\epsilon$-ratio-best-response* dynamics: Update the maximum-gain agent to its best response until product policy $\pi$ is $\epsilon$-ratio-Nash. Below, we make an assumption that also appears in the normal-form game setting and state the result. Our proof technique resembles its counterpart in the normal-form game setting, where a critical step is to use the MPG's smoothness and Assumption 2 to establish that any policy in the sequence, whether how suboptimal it is, will induce significant increase in the potential function, and therefore the number of highly suboptimal policies can be bounded because of the boundedness of the potential function.

**Assumption 2.** We have $0 < \Phi_\pi(s) \leq V_\pi(s)$ for any product policy $\pi$ and any state $s$.

**Theorem 4** (Proof in Appendix B.6). *Consider an $(\alpha, \beta)$-smooth MPG where Assumption 2 holds. Let $\sigma > 0$ be a constant for analysis. For the sequence of maximum-gain $\epsilon$-ratio-best-response policies $\pi_0, \ldots, \pi_T$, the number of policies therein that are not $\frac{\alpha}{(1+\beta)(1+\sigma)}$-optimal is at most*

$$\log_\rho \left(\Phi_{\max}/\Phi_0\right) - T \log_\rho \left(1/(1-\epsilon)\right) \tag{10}$$

*where $\rho = (1-\epsilon)(1 + \sigma(1+\beta)/N)$ and $\Phi_0 := \Phi_{\pi_0}(\mu)$.*

As BR dynamics (5,6,7) is an instance of maximum-gain approximate BR, Theorem 4 directly leads to a corollary that establishes a suboptimality guarantee for it, with details in Appendix B.7.

Theorem 4 does not provided an improved suboptimality guarantee at convergence: as $\sigma \to 0$, the suboptimality coefficient $\frac{\alpha}{(1+\beta)(1+\sigma)} \to \frac{\alpha}{1+\beta}$, matching that from Theorem 3 at convergence ($\epsilon = 0$). However, while Theorem 3 states about near-optimality *at convergence*, Theorem 4 bounds the number of highly suboptimal *during the updates* by maximum-gain BR and therefore suggests faster than alternatives like SGA: if the number of highly suboptimal policies specified in Eq. equation 10 is small, then $\pi_t$ can converge to near-optimality well before the last iterate.

# 6 PRACTICAL ALGORITHMS

The theoretical results on BR dynamics for MPGs in Section 5 rely on restrictive assumptions, including tabular policy parameterization, exact value function computation and maximum-gain coordination that would require centralized planning, and full state observability. However, the round-robin BR equation 8 offers a decentralized approach, making it highly practical for real-world applications. Unlike centralized methods that require global coordination, round-robin BR ensures that each agent updates independently in a sequential manner, further enhancing scalability and feasibility in multi-agent systems. Our empirical study (Section 7) demonstrates how round-robin BR can be incorporated into decentralized algorithms like DDPG (Lillicrap et al., 2015) and DreamerV2 (Hafner et al., 2021), enabling partial observability, neural network parameterization, and sample-based online learning. By setting $K = 1$ for BR variants to ensure fair comparison with SGA-based algorithms, we show that even this extreme choice leads to significant performance improvements.

# 7 EXPERIMENTS

The theoretical results show that BR matches (but is not superior to) SGA in terms of near-optimality guarantees at convergence. This prompts us evaluate them empirically here. Our evaluation begins with the tabular softmax setting with exact gradients where we compare maximum-gain BR against SGA, in an effort to reveal the difference between them that the theories did not reveal. Our results show that BR achieves better cooperation in most but all scenarios, which are consistent with our theoretical results. For complex scenarios where exact gradients and maximum gains are infeasible, we use round-robin BR for sample-based decentralized deep MARL algorithms. These algorithms with SGA are believe to be ineffective for cooperation, and people haven been relying on extensive centralization (e.g., centralized critic) to remedy it. However, we observe significant gains by only making the minimal modification of BR-based updates.

**Environments.** We evaluate on four benchmarks below. The first two are fully cooperative smooth MPGs and the second two are in more challenging partially observable settings.

*Matrix game.* The two-player stateless matrix game, a classic normal-form game, involves a shared team reward (payoff matrix in Figure 1), making it a stateless MPG with smoothness $(0, \beta)$ for any $\beta \geq 0$ (proof in Appendix C.1).

*Coordination game.* We generalize stateful coordination games with global observability (Zhang et al., 2021) to $N = 2, 3, 5$ players. The state and action spaces are defined as $\mathcal{S} = \mathcal{S}^1 \times \cdots \times \mathcal{S}^N$ and $\mathcal{A} = \mathcal{A}^1 \times \cdots \times \mathcal{A}^N$, where local states and actions $\mathcal{S}^i, \mathcal{A}^i = \{0, 1\}$. The team reward, detailed in Appendix D.2, incentivizes agents to align their local states. Each agent's state transition is defined by $P(s^i = 0 \mid a^i = 0) = 1 - \epsilon$ and $P(s^i = 0 \mid a^i = 1) = \epsilon$, with $\epsilon = 0.1$. This is a $(\alpha, \beta)$-smooth MPG, satisfying $\alpha V_{\max} = (1 + \beta)V_{\min}$ (proof in Appendix C.2).

*MPE.* We use two MPE (Lowe et al., 2017) tasks: Cooperative Navigation ($N = 6$) and Predator and Prey ($N = 6$), with agent-specific reward modifications. In Cooperative Navigation, rewards depend on covering assigned landmarks and penalizing individual collisions. In Predator and Prey, rewards are based on each agent's distance to its nearest prey and the prey it captures. Cooperative Navigation becomes approximately an MPG when collision penalties are ignored, particularly after agents learn to avoid collisions.

*SMAC.* We consider three SMAC tasks (Samvelyan et al., 2019): the heterogeneous tasks 2s_vs_1sc and 2s3z_vs_2s3z, and the homogeneous task 3m, where a group of allied agents fights against enemy agents. All three selected tasks have team reward and thus are MPG instances.

**Baselines.** For matrix and coordination games, we use the tabular softmax policy in Eq. (9) with PG-SGA and NPG-SGA as baselines, alongside their maximum-gain BR variants, PG-BR and NPG-BR, where the best performed $K$ is selected from $\{1, 5, 10, 30, 50\}$. In complex environments, where exact gradients and maximum-gain BR are infeasible, we adopt round-robin BR with $K = 1$. For MPE, we use decentralized MADDPG (DecDDPG) (Lowe et al., 2017) and its round-robin BR variant, DecDDPG-BR. For SMAC, we extend DreamerV2 (Hafner et al., 2021) to DecDreamer for decentralized training and execution, with

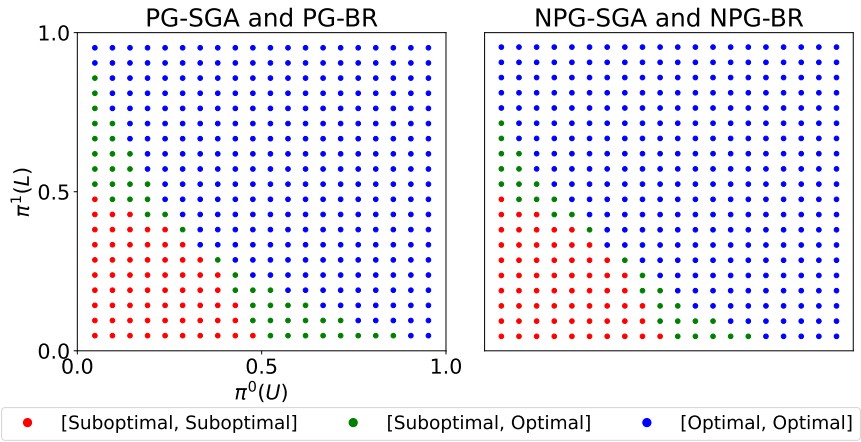

Figure 2: Each point represents the convergent policy types learned with SGA ($\pi^{\mathrm{SGA}}$) vs. BR ($\pi^{\mathrm{BR}}$), shown as $[\mathrm{type}(\pi^{\mathrm{SGA}}), \mathrm{type}(\pi^{\mathrm{BR}})]$, where the type of each convergent policy is either Optimal or Suboptimal (Nash). In the matrix game, 83.5% of initializations converge to optimal with both PG-BR ($K = 50$) and NPG-BR ($K = 50$), compared to 72.0% with PG and 76.0% with NPG. No points are [Optimal, Suboptimal].

Table 1: Optimal convergence rates (%) vs. $K$ (400 seeds).

|  | $K = 1$ | $K = 5$ | $K = 10$ | $K = 30$ | $K = 50$ |
|---|---|---|---|---|---|
| PG | 71.0% | 73.0% | 78.0% | 83.5% | 83.5% |
| NPG | 75.0% | 80.5% | 82.5% | 83.5% | 83.5% |

round-robin BR variant for world model updates denoted as DecDreamer-BR. The implementation details are in Appendix D.1.

**Results on Matrix Game.** The right side of Figure 1 starts with the initial policy parameters $\theta^0 = [\theta_U^0, \theta_D^0] = [0, 0]$ and $\theta^1 = [\theta_L^1, \theta_R^1] = [-2, 0]$. We compare the performance of PG-BR ($K = 50$) with the baseline PG, using exact gradients with different initial policies. Specifically, we divide the probability values of both $\pi_{\theta^0}^0(a^0 = U)$ and $\pi_{\theta^1}^1(a^1 = L)$ into 20 equally spaced intervals over the range $(0,1)$, resulting in 400 initial policies. Since each probability has only one degree of freedom, we fix both $\theta_D^0$ and $\theta_R^1$ at 0, so that $\theta_U^0$ and $\theta_L^1$ form a bijective map with the corresponding probabilities. As shown in Figure 2, PG-BR converges to the $\epsilon$-optimal policy for 83.5% of initial policies (blue and green points), outperforming the SGA's 72% (blue points), with $\epsilon = 0.001$. Similarly, NPG-BR achieves 83.5% convergence (blue and green points), surpassing the NPG-based SGA's 76% (blue points). Notably, with both PG and NPG, no initial policies lead to optimal SGA and suboptimal BR convergence, highlighting BR's advantage over SGA.

*Effect of K.* Table 1 shows the performance of (N)PG-BR in the matrix game with different $K$. The results confirm that larger $K$ leads to better joint policies.

**Results on Coordination Game.** Table 2 shows that when converging to $\epsilon$-Nash with $\epsilon = 0.1$, the POA for PG-BR is higher than that of PG for $N = 2$ and $N = 5$, but lower for $N = 3$. For NPG-BR, the POAs are consistently higher than those of NPG across all $N = 2, 3, 5$.

*Effect of K.* Figure 3 shows the best-response learning dynamics for different values of $K$. The results indicate that for PG-BR, larger $K$ leads to a higher POA for $N = 2$, consistent with the findings in the matrix game with also $N = 2$, while the POA remains relatively stable for larger numbers of agents $N = 3$ and $N = 5$. For NPG-BR, intermediate values of $K$ perform best: $K = 5$ yields the highest POAs for $N = 2$ and $N = 3$, while $K = 30$ is optimal for $N = 5$.

Table 2: POA on Coordination Game (mean$_{\text{std. err.}}$ over 10 seeds).

| $N$ | PG | NPG | PG-BR | NPG-BR |
|---|---|---|---|---|
| 2 | $0.91_{0.03}$ | $0.92_{0.03}$ | $0.92_{0.03}$ | $0.96_{0.02}$ |
| 3 | $0.89_{0.02}$ | $0.89_{0.02}$ | $0.87_{0.02}$ | $0.91_{0.02}$ |
| 5 | $0.88_{0.02}$ | $0.89_{0.02}$ | $0.89_{0.02}$ | $0.92_{0.02}$ |

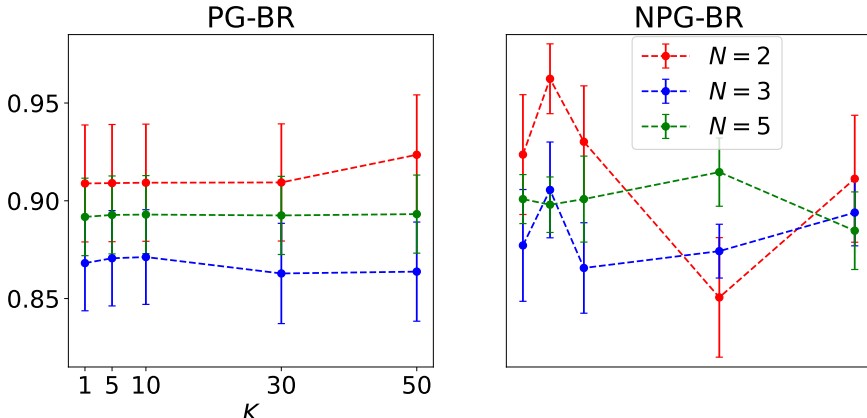

Figure 3: Effectiveness of $K$ in the coordination game.

**Results on MPE.** Figure 4(top) shows that DecDDPG performs better when learned with round-robin BR compared to SGA in both Cooperative Navigation ($N = 6$) and Predator and Prey ($N = 6$). This demonstrates that best-response dynamics can also be effective in more complex scenarios without exact gradient computation.

Figure 4(bottom) illustrates trajectories from policies learned via SGA and BR using 30% of training samples in Predator and Prey ($N = 6$), where episode rewards are similar but begin to differ. In DecDDPG, agents 3 and 4 and agents 1 and 6 overlap, indicating conflicts. DecDDPG-BR demonstrates improved cooperation: agents 1, 2, and 5 pursue Prey 1, while agents 2 and 4 focus on Prey 2.

**Results on SMAC.** Figure 5(top) shows that the model-based, decentralized, and sample-based algorithm DecDreamer-BR outperforms its SGA counterpart, DecDreamer. This holds true for the heterogeneous tasks 2s_vs_1sc and 2s3z_vs_2s3z, and the homogeneous task 3m. These results suggest that in decentralized settings, best-response dynamics can make the environment more stable, leading to better world model learning, which ultimately benefits model-based algorithms. As shown in the Figure 5(bottom), in 2s_vs_1sc, the policy learned by DecDreamer directs the allies to approach and attack the enemy with minimal movement, resulting in the loss of the controlled allies. In contrast, the policy learned by DecDreamer-BR leads the allies to attack the enemy with more dynamic movements, successfully defeating the enemy.

## 8 CONCLUSION

We motivated unilateral BR dynamics in MPGs and proved that the maximum-gain BR variant converges to a Nash equilibrium, with optimality guarantees in smooth MPGs. While the theoretical guarantees for BR largely match those for SGA, we empirically show the broad applicability and advantage of BR in improving cooperation in the classical domains of matrix games and coordination games, as well as in high-dimensional tasks solved by decentralized deep MARL. The theoretical results in this paper do not confirm that maximum-gain BR is superior to SGA in terms of improving cooperation at convergence. Note that our empirical results show that, in a few cases (e.g., coordination game with $N = 3$), SGA indeed finds better cooperation at convergence, and this suggests our theoretical claim is the best one can hope unless further

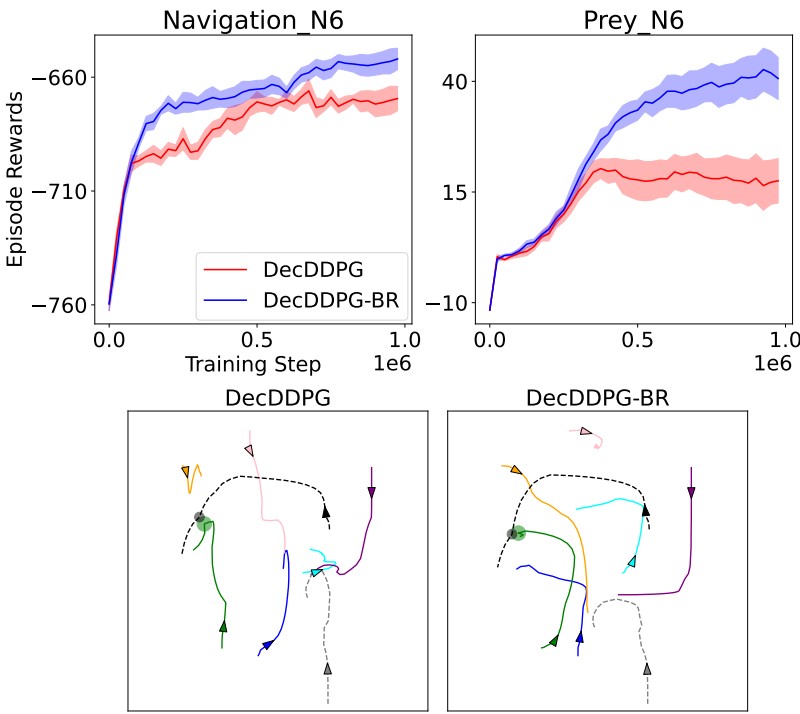

Figure 4: *Top:* Learning curves on MPE. *Bottom:* Example episodes at 30% training, with circles for captures; each agent is shown in a different color.

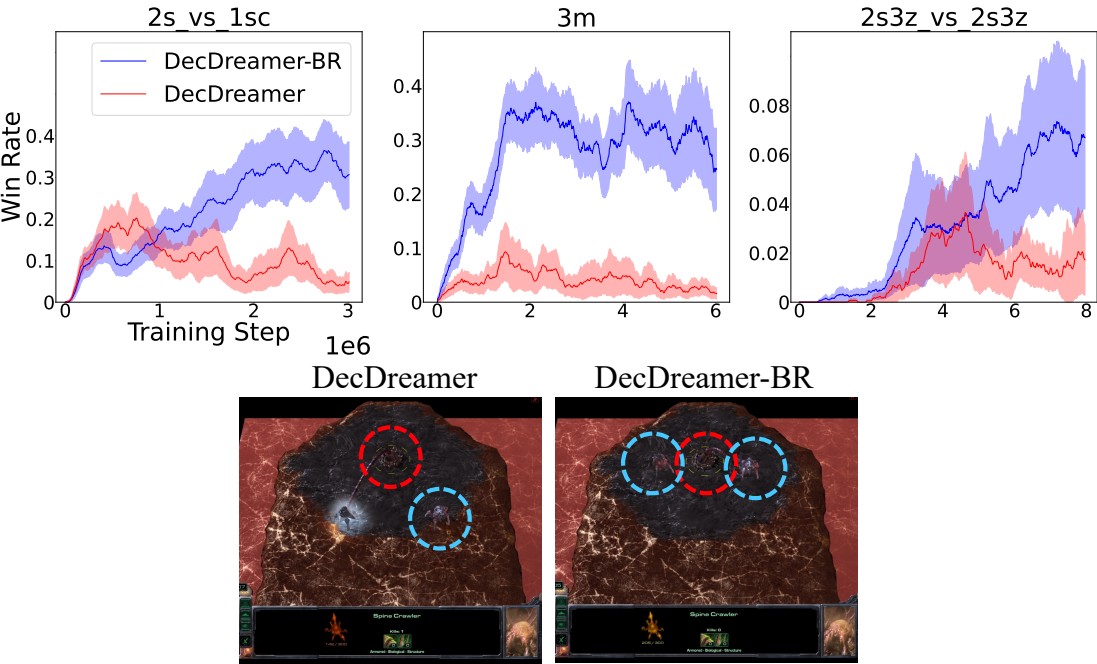

Figure 5: *Top:* Learning curves on SMAC. *Bottom:* Visualization of DecDreamer and DecDreamer-BR policies in 2s__vs__1sc. Allies and enemies are highlighted in blue and red dashed circles, respectively.

assumptions/conditions are made in favor of BR. Another future direction is to establish finite-time rates and suboptimality bounds for round-robin BR, as our current result gives only asymptotic convergence.

**Broader Impact**

This work aims to improve coordination in multi-agent systems, which may benefit applications such as robotics, traffic control, and distributed resource allocation. However, stronger multi-agent coordination can also have dual-use applications, and sequential adaptation mechanisms may behave unpredictably in real-world deployments due to partial observability, distribution shift, or unmodeled interactions. These considerations suggest that practical use should be accompanied by careful robustness testing, safety evaluation, and appropriate human oversight.

**Acknowledgment**

Xiaoling Zeng and Qi Zhang acknowledge funding support from National Science Foundation (NSF) award 2544947 and NSF CAREER award 2544948. Thinh T. Doan acknowledges funding support from the National Science Foundation (NSF) under CAREER Award 2527059 and AFOSR Grant FA9550-25-1-0247.

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

# A  Additional Related Work on BR Dynamics in Potential Games

Below we provide a detailed comparison with closely related works on BR and sequential-update dynamics, including Swenson et al. (2018); Marden (2012) in classical game theory, Maheshwari et al. (2022) in deep MARL, Song et al. (2021); Guo et al. (2023) in Markov game settings, and related uses of alternating updates in robust RL and adversarial training (Pinto et al., 2017; Zhang et al., 2022a).

Swenson et al. (2018); Marden (2012) consider BR dynamics with immediate optimal responses, whereas our work focuses on unilateral best-response dynamics, where a single agent adjusts its policy incrementally through a fixed number of gradient-based updates to approximate the best response. This approach is more practical to implement, especially in the deep MARL setting. Furthermore, Swenson et al. (2018) considers only stateless settings, whereas our work addresses stateful settings with state transitions. Marden (2012) studies state-based potential games, which are stateful with transitions and possess a potential structure. However, state-based potential games are a simplification of the broader class of Markov games by assuming a discount factor $\gamma = 0$, resulting in agents learning myopically to maximize utility at the current timestep. In contrast, our work considers Markov games with general discount factors. This distinction allows us to analyze Nash equilibria as responses to the value, or sum of discounted utilities, over future timesteps rather than only to current-stage utility.

Our work introduces a general framework (Lemma 1) that reduces MARL to single-agent reinforcement learning (RL) for analyzing unilateral BR dynamics. By contrast, Swenson et al. (2018) and Marden (2012) consider BR dynamics with immediate optimal responses and use classical game-theoretic tools, without establishing a connection to the single-agent RL setting.

Maheshwari et al. (2022) also consider Markov potential games (MPGs), as does our work. However, Maheshwari et al. (2022), while mentioning best-response dynamics for context and to relate its contributions to the broader literature, does not directly employ BR. Instead, it studies policy-gradient-based independent learning, focusing on decentralized updates and convergence to Nash equilibria without explicitly using unilateral BR dynamics.

Song et al. (2021) study Nash-CA for learning an $\epsilon$-approximate Nash equilibrium in finite-horizon general-sum Markov games, with sample complexity as the primary objective. In contrast, our paper studies discounted MPGs and analyzes unilateral approximate best response implemented by a fixed number $K$ of PG/NPG steps for the selected agent. Our main results concern convergence of this gradient-based update rule, welfare/suboptimality guarantees via smoothness and PoA-style analysis, and decentralized deep-MARL implementations. Guo et al. (2023) are closer in spirit. They study Markov $\alpha$-potential games, of which MPGs are the special case $\alpha = 0$, and analyze a Sequential Maximum Improvement Smoothed Best Response algorithm. Their method computes improvement scores over player–state pairs, selects the largest one, and updates only that player's policy at the selected state. In contrast, our method selects one agent and updates its full discounted policy through a finite inner loop of PG/NPG steps while fixing the other agents. Our analysis also covers both maximum-gain and round-robin selection and further develops welfare/suboptimality guarantees through smoothness and PoA-style analysis.

We also acknowledge that alternating or turn-taking updates are a broader optimization heuristic that appears beyond the BR literature. For example, robust adversarial RL methods such as Pinto et al. (2017) use alternating updates between a protagonist and an adversary, and alternating gradient methods have been studied extensively in minimax optimization (Zhang et al., 2022a). Our claim is therefore not that alternation itself is new in general. Rather, our contribution is to study this update structure in discounted MPGs, where freezing the other agents allows each unilateral update to be analyzed through a single-agent MDP lens, and where the resulting analysis connects not only to Nash convergence but also to cooperation quality through smoothness-based welfare guarantees.

Moreover, prior work by Swenson et al. (2018); Marden (2012); Maheshwari et al. (2022); Song et al. (2021); Guo et al. (2023) primarily studies convergence to Nash equilibria or efficient equilibrium learning, without our focus on quantifying the cooperation quality of the generated policies through smoothness and PoA-style guarantees in discounted MPGs. Our work develops these welfare-oriented guarantees together with practical BR-style implementations in deep MARL. Additionally, Swenson et al. (2018) assumes convexity

or monotonicity of the potential function to derive convergence-rate results, whereas Markov games involve nonconvex objectives, necessitating different analyses. Finally, Swenson et al. (2018) and Marden (2012) are purely theoretical and do not explore integration with deep MARL in complex environments, whereas our work studies this combination in settings such as MPE and SMAC.

## B   Proofs

### B.1   Proof of Lemma 1

Consider the $K$ iterations of Eq. equation 5 that begin at $t = lK$.

Case 1: $\theta_t$ is not an $\epsilon$-Nash policy, by definition there exists some $i$ such that its Nash-gap$^i(\theta_t) > \epsilon$. Because it is assumed that $i$ achieves at least $\frac{\epsilon}{2}$-best response, we have Nash-gap$^i(\tilde{\theta}_{t,K}^{i,-i}) \leq \frac{\epsilon}{2}$. The potential game structure in Eq. equation 1 therefore implies that the improvement in potential is at least $\frac{\epsilon}{2}$, i.e., $\Phi_{\tilde{\theta}_{t,K}^{i,-i}}(\mu) - \Phi_{\theta_t}(\mu) > \frac{\epsilon}{2}$. Since $i_l^*$ is chosen by the maximum-gain criterion in Eq. equation 7, the improvement in potential by selecting $i = i_l^*$ is also at least $\frac{\epsilon}{2}$, i.e., $\Phi_{\theta_{t+K}}(\mu) - \Phi_{\theta_t}(\mu) > \frac{\epsilon}{2}$.

Case 2: In the other case where $\theta_t$ is already an $\epsilon$-Nash policy, by the same reasoning above the improvement in potential is no larger than $\frac{\epsilon}{2}$, and therefore we can halt the updates at $t$ and return an $\epsilon$-Nash policy.

The above analysis implies that the potential is improved by at least $\frac{\epsilon}{2}$ every $K$ iterations until an $\epsilon$-Nash policy is found at $T$ with $T = O\left(\frac{\Phi_{\max} - \Phi_{\min}}{\epsilon^p}\right) \cdot \frac{2}{\epsilon}$.

### B.2   Proof of Theorem 1

The choice of $K$ guarantees $\frac{\epsilon}{2}$-best response with the $\eta$ required in the corresponding single-agent setting, and therefore the results follow directly from Lemma 1.

### B.3   Proof of Theorem 2

The proof first establishes the convergence of the potential function, which holds for both selection criteria, and then shows that this convergence implies a Nash policy in each case.

First, regardless of whether the agent is selected via maximum-gain or round-robin, each step involves a single agent updating its policy. This reduces the problem to a single-agent MDP. As per Lemma C.2 in Agarwal et al. (2019), a policy gradient update with a sufficiently small learning rate guarantees that the selected agent's value function increases monotonically. Due to the potential game structure, the total potential function $\Phi$ also increases monotonically. Given Assumption 2, which guarantees a bounded potential, the Monotone Convergence Theorem ensures that the sequence of potential function values $\{\Phi_{\theta_t}\}$ converges to a finite limit, $\Phi^*$.

The convergence of the potential function implies that the improvement gained at each update step must approach zero. We now show this leads to a Nash policy for each criterion:

*Maximum-Gain Criterion:* By definition, this criterion selects the agent offering the largest possible improvement at each step. Since this maximum improvement must go to zero for the potential function to converge, it follows that the potential improvement for *every* agent must also go to zero. This implies that as $t \to \infty$, $\forall i, \nabla_{\theta_t^i} \Phi_\theta(\mu) \to 0$.

*Round-Robin Criterion:* This criterion ensures every agent is updated infinitely often. Let $\{\theta_t\}$ have a limit point $\theta^*$. The vanishing improvements imply that the gradient of the agent being updated must be zero at the limit. Assume for contradiction that at $\theta^*$, the gradient for some agent $j$ is non-zero $(\nabla_{\theta^j} \Phi_{\theta^*}(\mu) \neq 0)$. By the continuity of the gradient, an update for agent $j$ in the neighborhood of $\theta^*$ would yield a non-trivial improvement. The round-robin schedule guarantees agent $j$ will be updated within this neighborhood, contradicting that all improvements must vanish. Therefore, the gradient for every agent must be zero at any limit point.

In both cases, the algorithm converges to a state where the gradient with respect to every agent's parameters is zero. For any agent $i$, the problem reduces to a single-agent MDP when other agents $-i$ are treated as part of the environment. By the single-agent result, Theorem 5.1 from Agarwal et al. (2019), when $\nabla_{\theta_t^i} \Phi_\theta(\mu) = 0$, agent $i$ behaves optimally in the reduced single-agent MDP, which means that $\forall i$, Nash-gap$^i(\pi^t) = 0$. Consequently, as $t \to \infty$, the joint policy $\pi^t$ converges to a Nash policy, as defined by the Nash equilibrium condition.

### B.4 Proposition 1

**Proposition 1** (Sufficient condition for smooth MGs). *The reward functions $\{r^i\}_{i \in \mathcal{N}}$ of a Markov game is said to be $(\lambda, \mu)$-smooth if $\lambda r_{\bar{\pi}}(s) \leq \sum_{i \in \mathcal{N}} r^i_{\bar{\pi}^i, \pi^{-i}}(s) \leq \mu r_\pi(s)$ for any state $s$ and any pair of product policies $\pi$ and $\bar{\pi}$, where $r^i_\pi(s) := \mathbb{E}_{a \sim \pi(s)}[r^i(s,a)]$, $r_\pi(s) := \sum_i r^i_\pi(s)$, and $(\lambda, \mu)$ are nonnegative. Letting $M_\pi := (I - \gamma P_\pi)^{-1}$, the transition function $P$ of a Markov game is said to be $(\kappa, \nu)$-smooth if $M_{\bar{\pi}^i, \pi^{-i}} r \geq \kappa M_{\bar{\pi}} r - \nu M_\pi r$ for any $r \in \mathbb{R}^{|\mathcal{S}|}$ and any pair of product policies $\pi, \bar{\pi}$ with nonnegative $(\kappa, \nu)$. For a Markov game, if its reward functions are $(\lambda, \mu)$-smooth and its transition function is $(\kappa, \nu)$-smooth, then the Markov game is $(\alpha = \kappa\lambda, \beta = \mu\nu)$-smooth.*

*Proof.* We can establish

$$
\begin{aligned}
\sum_i V^i_{\bar{\pi}^i, \pi^{-i}}(s) &= \sum_i M_{\bar{\pi}^i, \pi^{-i}} r^i_{\bar{\pi}^i, \pi^{-i}} \\
&\geq \sum_i \kappa M_{\bar{\pi}} r^i_{\bar{\pi}^i, \pi^{-i}} - \nu M_\pi r^i_{\bar{\pi}^i, \pi^{-i}} \\
&= \kappa M_{\bar{\pi}} \sum_i r^i_{\bar{\pi}^i, \pi^{-i}} - \nu M_\pi \sum_i r^i_{\bar{\pi}^i, \pi^{-i}} \\
&\geq \kappa M_{\bar{\pi}} \lambda r_{\bar{\pi}} - \mu M_\pi \mu r_\pi = \kappa\lambda V_{\bar{\pi}} - \mu\nu V_\pi
\end{aligned}
$$

where the two inequalities are due to the smoothness of the transition function and the reward functions, respectively, which completes the proof. $\qquad \square$

### B.5 Proof of Theorem 3

Consider setting $\bar{\pi} = \pi_*$ in Definition 4 where $\pi_*$ is a policy that achieves the optimal joint value. We have

$$
\sum_i V^i_\pi(s) \geq \sum_i (1 - \epsilon) V^i_{\pi^i_*, \pi^{-i}}(s) \geq (1 - \epsilon)\left(\alpha V_{\pi_*}(s) - \beta V_\pi(s)\right)
$$

where the first inequality is by the definition of $\pi$ being $\epsilon$-ratio-Nash and the second inequality is by the definition of smooth Markov game. Rearranging the terms completes the proof.

### B.6 Proof of Theorem 4

We abbreviate $V^i_\pi(\mu)$ as $V^i_\pi$ and $V_\pi(\mu)$ as $V_\pi$. For any policy $\pi_t$, define $\delta^i(\pi_t) := V^i_{\pi^i_*, \pi_t^{-i}} - V^i_{\pi_t}$ and $\Delta(\pi_t) := \sum_i \delta^i(\pi_t)$. We now have

$$
V_{\pi_t} = \sum_i V^i_{\pi_t} = \sum_i \left( V^i_{\pi^i_*, \pi_t^{-i}} - \delta^i(\pi_t) \right) \geq \alpha V_{\pi_*} - \beta V_{\pi_t} - \Delta(\pi_t)
$$

where the inequality is due to the $(\alpha, \beta)$-smoothness of the MPG, which implies

$$
V_{\pi_t} \geq \frac{\alpha}{1 + \beta} V_{\pi_*} - \frac{1}{1 + \beta} \Delta(\pi_t). \tag{11}
$$

For a policy $\pi_t$ that is not $\frac{\alpha}{(1+\beta)(1+\sigma)}$-optimal, we have

$$
\begin{aligned}
\Delta(\pi_t) \geq \alpha V_{\pi_*} - (1 + \beta) V_{\pi_t} &> (1 + \beta)(1 + \sigma) V_{\pi_t} - (1 + \beta) V_{\pi_t} \\
&= \sigma(1 + \beta) V_{\pi_t} \geq \sigma(1 + \beta)\Phi_t
\end{aligned}
$$

where the first inequality is directly from inequality equation 11, the second inequality due to that $\pi_t$ is a bad policy, the third due to the assumption that $\Phi_\pi(s) \leq V_\pi(s)$. Therefore, for the maximum-gain agent chosen to update from $t$ to $t + 1$, the increase in its local value is at least $\frac{\sigma(1+\beta)}{N}\Phi_t$ since $\Delta(\pi_t) = \sum_i \delta^i(\pi_t)$. Due to the characteristic of $\Phi$ in Eq. equation 1, we have $\Phi_{t+1} - \Phi_t \geq \frac{\sigma(1+\beta)}{N}\Phi_t$, i.e.,

$$
\Phi_{t+1} \geq (1 + \sigma(1 + \beta)/N)\,\Phi_t. \tag{12}
$$

After updating a $\pi_t$ that is $\frac{\alpha}{(1+\beta)(1+\sigma)}$-optimal, $\Phi$ increases by a ratio of at least $\frac{1}{1-\epsilon}$:

$$\frac{\Phi_{t+1} - \Phi_t}{\Phi_t} = \frac{V^i_{\pi_{t+1}} - V^i_{\pi_t}}{\Phi_t} = \frac{V_{\pi_{t+1}} - V_{\pi_t}}{\Phi_t} \geq \frac{V_{\pi_{t+1}} - V_{\pi_t}}{V_{\pi_t}} > \frac{1}{1-\epsilon} - 1.$$

Let $m$ and $T - m$ be the number of bad and good policies in the sequence, respectively. We then have $\Phi_0 \left(1 + \frac{\sigma(1+\beta)}{N}\right)^m \left(\frac{1}{1-\epsilon}\right)^{T-m} \leq \Phi_{\max}$, which implies equation 10 and concludes the proof.

## B.7 Corollary 1

**Corollary 1** (Suboptimality bound of maximum-gain approximate (N)PG-BR in smooth MPGs)**.** *Consider a $(\alpha, \beta)$-smooth MPG where Assumption 2 holds. Let $\pi_*$ be an optimal policy and $\sigma > 0$ be a constant for analysis. For the sequence of policies $\pi_0, \ldots, \pi_T$ generated from the maximum-gain approximate BR with Eq. (5,6,7) with PG update equation 3 with $K \geq \frac{2}{c(1-\gamma)^6\epsilon}$ or with NPG update equation 4 with $K \geq \frac{4}{(1-\gamma)^2\epsilon}$. the number of policies therein that are not $\frac{\alpha}{(1+\beta)(1+\sigma)}$-optimal is at most*

$$\log_\rho \left(\Phi_{\max}/\Phi_0\right) - T \log_\rho \left(1 + \frac{\epsilon}{2(1-\gamma)}\right) \tag{13}$$

*where $\rho = \left(1 + \sigma(1+\beta)/N\right)\big/\left(1 + \frac{\epsilon}{2(1-\gamma)}\right)$.*

*Proof.* Similar to the proof of Theorem 4, we can obtain inequality equation 12 for a bad $\pi_t$, and for a good $\pi_t$, the $\epsilon/2$ increase per iteration, which is guaranteed by the choice of $K$, implies

$$\frac{\Phi_{t+1} - \Phi_t}{\Phi_t} = \frac{V^i_{\pi_{t+1}} - V^i_{\pi_t}}{\Phi_t} = \frac{V_{\pi_{t+1}} - V_{\pi_t}}{\Phi_t} \geq \frac{V_{\pi_{t+1}} - V_{\pi_t}}{V_{\pi_t}} > \frac{\epsilon/2}{1-\gamma}.$$

Let $m$ and $T - m$ be the number of bad and good policies in the sequence, respectively. We then have

$$\Phi_0 \left(1 + \frac{\sigma(1+\beta)}{N}\right)^m \left(1 + \frac{\epsilon}{2(1-\gamma)}\right)^{T-m} \leq \Phi_{\max},$$

which implies equation 13 and concludes the proof. $\square$

## C  Smoothness of Toy Examples

### C.1  The Matrix Game

To satisfy the smoothness condition in definition (4), it is sufficient to have

$$V_{\min} + V_{\min} \geq \alpha(V_{\max} + V_{\max}) - \beta(V_{\min} + V_{\min})$$

$$\longrightarrow 2V_{\min} \geq 2\alpha V_{\max}$$

$$\longrightarrow \alpha \leq \frac{V_{\min}}{V_{\max}} = 0$$

Since $\alpha$ is nonnegative, we can have $\alpha \geq 0$ as a sufficient condition for the smoothness of this MPG.

### C.2  The Coordination Game

By Proposition 1, to prove Coordination Game is a smooth Markov game, it is sufficient to show the transition and reward smoothness.

#### C.2.1  Reward smoothness in the coordination game.

In Coordination Game, we know that all agents share a team reward which only depends on state, i.e.,

$$\forall i, j, s, a, a\prime, r^i(s, a) = r^j(s, a\prime)$$
$$\longrightarrow \forall s, \pi, \pi\prime, r_{\pi\prime}(s) = r_\pi(s) = \sum_i r_\pi^i(s)$$

To satisfy $\forall s, \pi, \pi\prime, \lambda r_{\pi\prime}(s) \leq \sum_i r_{\pi\prime^i, \pi^{-i}}^i(s) \leq \mu r_\pi(s)$, it suffices to have

$$\lambda r_\pi(s) \leq \sum_i r_\pi^i(s) \leq \mu r_\pi(s) \iff \lambda \leq 1 \leq \mu \tag{14}$$

#### C.2.2  Transition smoothness in the coordination game

To have $\forall s, \pi, \pi\prime, M_{\pi\prime^i, \pi^{-i}} r \geq \kappa M_{\pi\prime} r - \nu M_\pi r$, it suffices to have

$$V_{\min} \geq \kappa V_{\max} - \nu V_{\min} \longrightarrow \frac{\kappa}{1+\nu} \leq \frac{V_{\min}}{V_{\max}} \tag{15}$$

, where $V_{\max}$ and $V_{\min}$ are the largest and smallest values given any joint policy in Coordination Game.

Coordination game is therefore $(\alpha = \kappa\lambda, \beta = \mu\nu)$-smooth, with constraints (14) and (15). For the suboptimality bound of 0-ratio-Nash policy below, we choose the $\lambda = \mu = 1$ and $\frac{\kappa}{1+\nu} = \frac{V_{\min}}{V_{\max}}$ which satisfy the constraints. Therefore, it is a $(\alpha, \beta)$-smooth MPG for any $\alpha, \beta$ satisfy $\alpha V_{\max} = (1 + \beta)V_{\min}$.

# D Experiment Details

## D.1 Implementation Details

For decentralized DDPG, each agent parameterizes its policy and a critic that approximates its action-value function, both with neural networks and from its partial observations. Between update iterations, all agents take action to finish an episode, after which each agent updates its own replay buffer that stores recent transitions. Within each iteration, an agent is selected based on the round-robin criterion equation 8, and only that agent updates its parameters by sampling transitions from its replay buffer and taking the gradient step, following standard DDPG practice.

For decentralized DreamerV2, which is a model-based algorithm, each agent in addition parameterizes and learns a world model (from its local observations and actions).Each agent has its own replay buffer and round-robin BR is similarly integrated into the DreamerV2 practice.

## D.2 Pseudocode for the Reward Function of Coordination Game

---
**Algorithm 1** Calculate the team reward for $N$ agents in state $s$

---
1: **if** $(N = 2)$ or $(N = 3)$ **then**
2:     difference_bound = 1
3: **else**
4:     difference_bound = 2
5: **end if**
6: **if** $abs(s.\text{count}("0") - s.\text{count}("1")) \leq$ difference_bound **then**
7:     **if** $s.\text{count}("0") < s.\text{count}("1")$ **then**
8:         reward $= 1$
9:     **else**
10:         reward $= 0$
11:     **end if**
12: **else if** $s.\text{count}("0") > s.\text{count}("1")$ **then**
13:     reward $= 3$
14: **else**
15:     reward $= 2$
16: **end if**

---

### D.3 Hyperparameters

Table 3: Hyperparameters for the Matrix Game

| Hyperparameter | Value |
|---|---|
| $\gamma$ (discount factor) | 0 |
| $\eta$ | 0.01 |

Table 4: Hyperparameters for Coordination Game

| Hyperparameter | Value |
|---|---|
| $\gamma$ (discount factor) | 0.95 |
| $\mu$ (initial state distribution) | Uniform |
| $\epsilon$ | 0.1 |
| $\eta$ | 0.01 |
| #seeds | 10 |

Table 5: Hyperparameters for MPE

| Hyperparameter | Value |
|---|---|
| Episode length | 25 |
| Number of training episodes | 40000 |
| Discount factor | 0.95 |
| Batch size from replay buffer | 1024 |
| Actor's learning rate | 1e-4 |
| Actor network architecture | $o^i$-FC(128)-ReLU-FC(128)-ReLU-FC($|\mathcal{A}^i|$) |
| Critic's learning rate | 1e-3 |
| Critic network architecture | Concat($o^i, a^i$)-FC(128)-ReLU-FC(128)-ReLU-FC(1) |
| Optimizer for both actor and critic | Adam |
| #episodes per evaluation | 200 |
| #seeds | 5 |

$o^i$ is the local observation for agent $i$.

Table 6: Hyperparameters for SMAC

| Hyperparameter | Value |
| --- | --- |
| Number of training epochs for the model | 5 |
| Number of training epochs for the actor and critic | 4 |
| Number of sampled rollouts | 40 |
| Sequence Length | 20 |
| Rollout Length | 15 |
| Buffer capacity | 250000 |
| Batch size | 2000 |
| Model's learning rate | 3e-4 |
| Actor's learning rate | 1e-6 |
| Actor network architecture | State[i]-FC(256)-Normalize-ReLU-FC(256)-Normalize-ReLU-FC(9)- Normalize-OneHotCategorical |
| Critic's learning rate | 1e-4 |
| Critic network architecture | State[i]-FC(256)-Normalize-ELU-FC(256)-Normalize-ELU-FC(1)-Normalize |
| Optimizer for both actor and critic | Adam |
| Entropy coefficient | 0.01 |
| Entropy annealing | 0.99998 |
| Gradient clipping norm | 20 |
| #seeds | 10 |

State[i] contains the latent state and embedding for agent $i$. Hyperparameters not listed are consistent with those in Venugopal et al. (2023).

### D.4   Computing Resources

We implement our code in PyTorch and run our experiments on NVIDIA Tesla V100 GPUs and NVIDIA A100 GPUs.

