# OpenReview forum: "Analyzing Best-Response Dynamics for Cooperation in Markov Potential Games"
_TMLR — Accepted by TMLR_

### Review · Reviewer_ntNv · 2026-02-21

**Summary Of Contributions:**

This paper investigates unilateral Best-Response (BR) dynamics as a solution to non-stationarity and suboptimality in Multi-Agent Reinforcement Learning (MARL). While Simultaneous Gradient Ascent (SGA) creates a moving target problem through concurrent updates , the authors propose a turn-taking approach where only one agent updates at a time while others remain fixed. This provides a unified framework that reduces MARL to a sequence of single-agent problems , allowing the authors to establish the first finite-time convergence rates for approximate BR in Markov Potential Games (MPGs). By extending smoothness and Price of Anarchy to MPGs, they provide suboptimality guarantees that prove BR can achieve near-optimal cooperation. The work further ensures decentralized scalability by proving the asymptotic convergence of the practical Round-Robin criterion. Finally, empirical validation across high-dimensional tasks like StarCraft (SMAC) and MPE demonstrates that BR significantly outperforms standard simultaneous update methods.

## Strengths
* Theoretical Rigor: The paper provides a solid mathematical foundation for a common practitioner's trick (alternating training), offering specific convergence rates and suboptimality bounds that were previously missing for MPGs.
* Practicality: The integration of BR into decentralized deep RL algorithms (like DecDDPG and DecDreamer) shows that the method is not just a theoretical curiosity but scales to complex, modern benchmarks.
* Clarity of Benefit: The use of toy examples (the 2-player matrix game) clearly illustrates why BR avoids the suboptimal "traps" that simultaneous updates frequently fall into.

## Weaknesses
* The Theory-Practice Gap for K: The theoretical convergence rates often require a large number of inner-loop iterations (K), but the best empirical results in complex environments were achieved with K=1. This discrepancy is not fully reconciled.
* Incremental Novelty of Unilateral Updates: The concept of unilateral or "turn-taking" updates is fundamentally similar to alternating training or coordinate ascent/descent used in robust RL and adversarial training. While the application to MPGs is valuable, the structural approach of freezing other agents to maintain stationarity is a well-established heuristic in the broader optimization and machine learning literature.
* Strict Smoothness Assumptions: The suboptimality guarantees (Price of Anarchy) rely on the "smoothness" condition (Definition 4) and Assumption 3. Verifying these conditions in complex, real-world environments is non-trivial, potentially limiting the utility of these bounds for practitioners.

**Audience:**

Yes

**Audience Explanation:**

Yes. The findings of this paper would be of significant interest to the TMLR audience, particularly those focused on multi-agent reinforcement learning (MARL), decentralized optimization, and game theory.

**Claims And Evidence:**

Yes

**Claims Explanation:**

The submission’s claims are supported by accurate and clear evidence through a combination of rigorous mathematical proofs and extensive empirical testing. Theoretically, the authors successfully establish finite-time convergence rates for approximate Best-Response (BR) in Markov Potential Games (MPGs), matching existing Simultaneous Gradient Ascent (SGA) benchmarks while proving that unilateral updates effectively reduce non-stationarity by treating the process as a sequence of stationary single-agent problems. The introduction of "smoothness" and "Price of Anarchy" to the Markov setting provides a convincing basis for why BR finds near-optimal cooperation, with Theorem 4 offering a unique guarantee that bounds the number of suboptimal policies during updates. Empirically, the evidence is consistent across scales, demonstrating significant performance gains in decentralized settings. Although a gap exists where empirical success with K=1 exceeds what the large-K finite-time theory strictly requires, the overall results clearly demonstrate the benefits of BR.

**Requested Changes:**

* Address the Theory-Practice Discrepancy for K: Theorem 1 requires a large number of inner-loop updates (K) to guarantee convergence rates , yet the best empirical results on complex tasks (SMAC and MPE) were achieved with K=1. Please provide a discussion or a preliminary theoretical intuition explaining why K=1 performs so well despite being a significant departure from the theory’s requirements.
* Expand on the "Alternating Training" Connection: In the Related Work section, it would be beneficial to explicitly compare unilateral BR dynamics with the "alternating training" or "inner-outer loop" optimization used in robust RL and adversarial training. Acknowledging this structural similarity while highlighting your unique contributions to cooperative MPGs would better contextualize the work for the TMLR audience.

---

> ### Author Response · Authors · 2026-03-12
>
> Thank you for your review!
> Below is our response and we have revised the pdf to incorporate the Requested Changes.
> We will be happy to engage in further discussion.
>
> **Requested Change 1**
> We make following clarifications on $K$:
> - Our Theorem 1 establishes a coverage rate for large $K$, while Theorem 2 establishes asymptotic convergence for any fixed $K$. Therefore, our theory does not rule out the effectiveness of small $K$; rather, we currently do not know how to prove a comparable finite-time rate for small $K$.
> - In the deep-MARL experiments, we use $K=1$ because this matches the standard practice that uses the SGA baseline, which ensures fair and meaningful comparison.
> - We did try larger $K$ in MPE and observed performance similar to $K=1$, while both outperformed SGA; we speculate that this is because the Adam optimizer uses momentum, which can make repeated single-step updates with $K=1$ behave similarly to using a larger $K$.
>
> **Requested Change 2**
> Appendix A is revised to include more related works on alternating training, namely [1, 2].
> Specifically, robust adversarial RL methods such as RARL [1] train a protagonist and an adversary in an alternating zero-sum manner, and the broader minimax-optimization literature has also studied alternating updates extensively [2].
> We will be happy to incorporate more works if suggested.
>
> We do not claim “turn-taking” itself as our novelty. Rather, our contribution is to show that this update structure is particularly useful to induce cooperation in Markov potential games. We have revised Section 2 to clarify this distinction and avoid overstating the novelty of unilateral updates as a generic heuristic.
>
> ### References
> [1] Lerrel Pinto, James Davidson, Rahul Sukthankar, and Abhinav Gupta. Robust
> adversarial reinforcement learning. In International conference on machine
> learning, pages 2817–2826. PMLR, 2017.
>
> [2] Guodong Zhang, Yuanhao Wang, Laurent Lessard, and Roger B Grosse. Near-
> optimal local convergence of alternating gradient descent-ascent for mini-
> max optimization. In International Conference on Artificial Intelligence and
> Statistics, pages 7659–7679. PMLR, 2022.

---

### Review · Reviewer_8jYd · 2026-02-24

**Summary Of Contributions:**

The authors study two variants of sequential best response update in markov potential games. Moreover they also study the price of anarchy of these updates using the notions of smoothness. They also empirically study the performance of these algorithms in comparison to simultaneous update.

**Additional Comments:**

None

**Audience:**

No

**Audience Explanation:**

I find the contirbution of this submission to be marginal and not of great interest to TMLR readers.

**Claims And Evidence:**

No

**Claims Explanation:**

The paper studies a very common algorithm i.e. sequential best response in a very common setup of Markov potenital games. They do not cite some prior work which also study sequential best repsonse like:

-- For example NashCA algorithm in:  Song Z, Mei S, Bai Y. When can we learn general-sum Markov games with a large number of players sample-efficiently?. arXiv preprint arXiv:2110.04184. 2021 Oct 8.


-- For example sequential maximum improvement algorithm in: Guo X, Li X, Maheshwari C, Sastry S, Wu M. Markov $\alpha $-Potential Games. IEEE Transactions on Automatic Control. 2025 Jul 15.

Moreover, they also study the price of anarchy in Markov games and do not cite another very closely related paper
-- Zhang R, Zhang Y, Konda R, Ferguson B, Marden J, Li N. Markov games with decoupled dynamics: Price of anarchy and sample complexity. In2023 62nd IEEE Conference on Decision and Control (CDC) 2023 Dec 13 (pp. 8100-8107). IEEE.

The resutls in the paper, in my opinion, are standard and not very surprising. Therefore, I think this paper has marginal contribution.

**Requested Changes:**

-- If authors consider tabular setting then Assumption 2 is not needed and hold automatically.

-- Following closely related paper must be cited and the contributions should be compared
1. Song Z, Mei S, Bai Y. When can we learn general-sum Markov games with a large number of players sample-efficiently?. arXiv preprint arXiv:2110.04184. 2021 Oct 8.
2. Zhang R, Zhang Y, Konda R, Ferguson B, Marden J, Li N. Markov games with decoupled dynamics: Price of anarchy and sample complexity. In2023 62nd IEEE Conference on Decision and Control (CDC) 2023 Dec 13 (pp. 8100-8107). IEEE.
3. Guo X, Li X, Maheshwari C, Sastry S, Wu M. Markov $\alpha $-Potential Games. IEEE Transactions on Automatic Control. 2025 Jul 15.

---

> ### Author Response · Authors · 2026-03-12
>
> **Weakness 1: Comparison with Song et al. and Guo et al.**
> We have added this discussion in Appendix B. Song et al. study Nash-CA for learning an $\epsilon$-approximate Nash equilibrium in finite-horizon Markov games, with sample complexity as the primary objective. The present paper instead studies discounted MPGs and analyzes the unilateral approximate best response implemented by a fixed number $K$ of PG/NPG steps for the selected agent. The main results concern convergence of this gradient-based update rule, welfare/suboptimality guarantees via smoothness and PoA-style analysis, and decentralized deep-MARL implementations.
>
> Guo et al. are closer in spirit. They study Markov $\alpha$-potential games, of which MPGs are the special case $\alpha=0$, and analyze a Sequential Maximum Improvement Smoothed Best Response algorithm. Their method computes improvement scores over player--state pairs, selects the largest one, and updates only that player's policy at the selected state. In contrast, the present method selects one agent and updates its full discounted policy through a finite inner loop of PG/NPG steps while fixing the others. The analysis also covers both maximum-gain and round-robin selection, and further develops welfare/suboptimality guarantees through smoothness and PoA-style analysis. These papers are now cited and discussed explicitly.
>
> **Weakness 2: Comparison with Zhang et al.**
> We have added this discussion to Section 2 (Related Work) in the main text. The main connection is on the price-of-anarchy side. Zhang et al. study finite-horizon Markov games with decoupled dynamics under local policies, introduce a notion of smooth Markov games, and use it to bound the price of anarchy. For the MPG case, they also develop a distributed multi-agent soft policy iteration algorithm with Nash-convergence and sample-complexity guarantees. In contrast, the present paper studies discounted MPGs and uses smoothness not only to relate near-Nash policies to near-optimality, but also to derive a suboptimality bound along the maximum-gain BR trajectory. Their PoA result is therefore a game-level equilibrium-quality guarantee in a finite-horizon decoupled-dynamics setting, whereas the present smoothness analysis also gives guarantees for the policies generated by the learning dynamics. This comparison is now included in the revised related-work section.
>
> **Requested Change 1**
> Indeed, Assumption 2 is automatic for tabular settings.
> We’ve revised the pdf to reflect this.
> Thank you!
>
> **Requested Change 2**
> The suggested papers [1-3] are now cited and compared in the revised pdf: [2] is discussed in the related-work section of the main text, while [1] and [3] are discussed in Appendix B with detailed comparison to our setting and contributions.

---

### Review · Reviewer_ifby · 2026-02-27

**Summary Of Contributions:**

The paper proposes replacing simultaneous policy updates with an approximate unilateral best-response (BR) dynamic, where one agent updates while others are fixed. Agents are selected either by maximum potential improvement or via a practical round-robin rule.

The authors:
- show convergence guarantees for policy-gradient-based approximate BR in Markov potential games;
- extend smoothness arguments to relate Nash policies to social welfare;
- derive bounds on how many highly sub-optimal policies may appear along BR learning trajectories;
- demonstrate empirically that round-robin BR improves cooperation and performance in decentralized deep MARL settings.

Key strengths:
- Simple and intuitive idea that is easy to implement.
- Nice connection between MARL learning dynamics and price-of-anarchy (PoA) / smoothness theory.
- Empirical evidence suggests unilateral updates can bias learning toward better equilibria.
- Bridges theory and practical deep MARL implementations.

Key weaknesses:
- The strongest theoretical guarantees apply to maximum-gain BR, while experiments use round-robin BR;
- Some convergence arguments lack full technical precision;
- I believe that experimental comparisons are somewhat limited;
- Certain assumptions (for example, relation between potential and welfare) are insufficiently motivated in my view.

**Audience:**

Yes

**Audience Explanation:**

Researchers working on MARL theory, decentralized learning, or equilibrium selection would likely find the results interesting. The idea that update scheduling alone can influence equilibrium quality is both intuitive and practically relevant, and the smoothness-based welfare perspective should appeal to part of the TMLR audience.

**Broader Impact Concerns:**

I believe a brief Broader Impact discussion could be added, but I do not see this as fundamental.

For example, consider the following:
- Improved multi-agent coordination may have dual-use applications,
- Sequential adaptation mechanisms could behave unpredictably in real multi-agent deployments,

A short acknowledgement of these aspects would suffice.

**Claims And Evidence:**

Yes

**Claims Explanation:**

The empirical evidence supports the central intuition that unilateral updates can improve cooperation and stability. However, I believe the theoretical claims would benefit from tighter statements and clearer assumptions, particularly regarding convergence under round-robin updates.

In addition, some empirical claims appear stronger than what the presented comparisons fully justify. Overall, the evidence is promising but not entirely conclusive yet. Looking forward to clarifications from the authors.

**Requested Changes:**

Most relevant:
1. Strengthen theoretical rigor. The convergence argument (especially Theorem 2) should clearly specify assumptions, optimization conditions, and the precise single-agent results being invoked.
2. Clarify PoA and evaluation metrics. The definition and computation of cooperation / welfare metrics should be explicit in the main text.
3. Address theory–practice mismatch. The main welfare guarantees apply to maximum-gain BR, whereas experiments use round-robin updates. This gap should either be theoretically discussed or empirically justified.
4. Moderate strong empirical claims. Statements about achieving unprecedented decentralized performance should be softened or supported with broader baselines (ref: last sentence on the first paragraph of experiments section).

Other suggestions:
- Add stronger MARL baselines and clearer compute fairness reporting.
- Provide ablations over number of BR steps (K) and scheduling choices.
- Better explain when key assumptions (e.g., Assumption 3) hold.
- Improve discussion of how smoothness conditions relate to practical environments.

---

> ### Author Response · Authors · 2026-03-12
>
> Thank you for your review!
> Below is our response and we have revised the pdf to incorporate the Requested Changes.
> We will be happy to engage in further discussion.
>
> **Requested Change 1**
> To strengthen theoretical rigor, we have revised the pdf to with the following changes:
> - We have removed Assumption 2, because it holds automatically (as noted by Reviewer 8jYd). We have then explicitly stated that Assumption 1 is assumed throughout Section 5. Theorem 4 requires an additional assumption (originally Assumption 3, now Assumption 2), which was already stated in Theorem 4 explicitly.
>
> We believe no more changes are needed for theoretical rigor:
> - The optimization conditions (such as learning rate, number of iterations) was already stated clearly.
> - The precise single-agent results being invoked is more appropriately stated in the proof rather than in the theorem statement. Accordingly, in the appendix, the proof explicitly identifies the result we use, Theorem 5.1 of Agarwal et al. (2019), which is applied after reducing each unilateral update to a single-agent MDP with the other agents' policies fixed.
>
> **Requested Change 2**
> We formulated our cooperation/welfare metric in Definition 3 as a ratio regarding values summed over all agents, with text preceding it as the motivation. We intentionally avoided the term “price of anarchy” / “POA” (except when mentioning related works), because by convention POA is inversely proportional to the ratio we use  in Definition 3.
>
> Therefore, we believe no major changes are needed to formalize our cooperation/welfare metric. But we are open to further suggestions.
>
> **Requested Change 3**
> The tabular experiments (Matrix and Coordination games) explicitly use maximum-gain BR, in the setting identical to that for our theoretical results.
> In contrast, the deep-MARL experiments (MPE, SMAC) use round-robin BR because exact maximum-gain selection is impractical there: it requires centralized gain evaluation, accurate per-agent value estimation, tight synchronization, and full observability. The distinction is therefore intentional.
>
>
> **Requested Change 4**
> Per suggestion, we have removed the last sentence of the first paragraph of the experiments in the revised pdf.
>
> **Broader Impact Concerns**
> We have included a broader impact section in the revised pdf.

---

> > ### Comment · Reviewer_ifby · 2026-04-03
> > **Response to authors**
> >
> > Thank you for the detailed response and revisions. The paper is stronger after rebuttal: the related-work discussion is improved, some empirical overstatement has been removed, and several of my presentation-level concerns were addressed. My main substantive reservations are only partially resolved, however, in particular the precision of the asymptotic convergence argument and the remaining gap between the strongest theory and the practical round-robin setting. Overall, the response moved me in a positive direction, though not to an unqualified endorsement. Thank you again.

---

### Decision · Action_Editor_9QYs · 2026-04-20

**Recommendation:** Accept as is

**Audience:**

Yes

**Audience Explanation:**

This work is of interest to TMLR community due to its contribution to multi-agent reinforcement learning, which is an active area of research in the field. All the reviewers agree on this point.

**Claims And Evidence:**

Yes

**Claims Explanation:**

This work analyzes the dynamics of unilateral best-response (BR) updates for Markov Potential Games. The authors prove finite-time guarantees under maximum gain criterion which selects the agent that would maximally improve the total potential. The authors then show asymptotic guarantees with round-robin selection which updates the agents in a cyclic manner. The authors then conduct deep MARL experiments to corroborate their theory. Reviewers ifby and ntNv are in agreement for acceptance because the authors improve the literature survey and theoretical rigour in the revisions. However, the mismatch between the theory and practice is noted. Still, these reviewers are in agreement that the claims are supported by sufficient evidence.

Reviewer 8jYd voted for rejection because they found the novelty marginal. However, this is not a criterion for acceptance at TMLR. This reviewer did not argue against the soundness of the claims. As a result, I vote to accept the paper.